# Tumor Infiltrating Regulatory T Cells in Sporadic and Colitis-Associated Colorectal Cancer: The Red Little Riding Hood and the Wolf

**DOI:** 10.3390/ijms21186744

**Published:** 2020-09-14

**Authors:** Massimo Claudio Fantini, Agnese Favale, Sara Onali, Federica Facciotti

**Affiliations:** 1Department of Medical Science and Public Health, University of Cagliari, 09042 Cagliari, Italy; agnesefavale@icloud.com; 2CEMAD-IBD UNIT-Unità Operativa Complessa di Medicina Interna e Gastroenterologia, Dipartimento di Scienze Mediche e Chirurgiche, Fondazione Policlinico Universitario “A. Gemelli” IRCCS, 00168 Rome, Italy; saraonali@yahoo.it; 3Department of Experimental Oncology, IEO European Institute of Oncology IRCCS, 20139 Milan, Italy; federica.facciotti@ieo.it

**Keywords:** colorectal cancer, colitis-associated colorectal cancer, regulatory T cells, inflammation

## Abstract

Regulatory T cells represent a class of specialized T lymphocytes that suppress unwanted immune responses and size the activation of the immune system whereby limiting collateral damages in tissues involved by inflammation. In cancer, the accumulation of Tregs is generally associated with poor prognosis. Many lines of evidence indicate that Tregs accumulation in the tumor microenvironment (TME) suppresses the immune response against tumor-associated antigens (TAA), thus promoting tumor progression in non-small cell lung carcinoma (NSLC), breast carcinoma and melanoma. In colorectal cancer (CRC) the effect of Tregs accumulation is debated. Some reports describe the association of high number of Tregs in CRC stroma with a better prognosis while others failed to find any association. These discordant results stem from the heterogeneity of the immune environment generated in CRC in which anticancer immune response may coexists with tumor promoting inflammation. Moreover, different subsets of Tregs have been identified that may exert different effects on cancer progression depending on tumor stage and their location within the tumor mass. Finally, Tregs phenotypic plasticity may be induced by cytokines released in the TME by dysplastic and other tumor-infiltrating cells thus affecting their functional role in the tumor. Here, we reviewed the recent literature about the role of Tregs in CRC and in colitis-associated colorectal cancer (CAC), where inflammation is the main driver of tumor initiation and progression. We tried to explain when and how Tregs can be considered to be the “good” or the “bad” in the colon carcinogenesis process on the basis of the available data concluding that the final effect of Tregs on sporadic CRC and CAC depends on their localization within the tumor, the subtype of Tregs involved and their phenotypic plasticity.

## 1. Introduction

Regulatory T cells (Tregs) represent a class of lymphocytes evolved to maintain host’s homeostasis by preventing the activation of the immune system towards self or harmless antigens [1]. However, Tregs are also believed to promote tumor development. Cells with suppressive capacity were initially identified in the 80s but only in 1995, Shimon Sakaguchi described the suppressive phenotype of CD4+ T cells characterized by the high expression of the IL2 receptor (CD25) [2]. Since then, the biology and immunological properties of Tregs have been extensively investigated in a wide spectrum of human diseases including cancer. Infiltration of Tregs in cancer tissue is associated with a worse prognosis in many cancers including non-small cells lung cancer (NSCLC), breast carcinoma and melanoma. Tregs are believed to accumulate in response to cytokines and chemokines released in the tumor microenvironment (TME) by dysplastic cells and tumor stroma infiltrating cells, to suppress the anti-tumor activity mediated by natural killer (NK) and cytotoxic CD8+ T cells and to favor tumor progression [3]. In sporadic colorectal cancer (CRC), the association between Tregs infiltration and prognosis is unclear. Indeed, while high frequency of Tregs among tumor infiltrating lymphocytes (TILs) have been correlated with tumor stage and shorter disease-free survival, in other studies, Tregs infiltration was associated with reduced tumor growth [4]. These discordant results might be, at least in part, reconciled, considering the heterogeneity of cells expressing FoxP3 (the transcription factor used to identify Tregs), their phenotypic plasticity and the role played by the unique bacteria-induced inflammatory environment existing in CRC. The role of Tregs in CRC is further complicated in the case of colitis-associated colorectal cancer (CAC) where chronic inflammation is the major driver of tumor initiation, promotion and progression [5]. Here, we first provide a general overview on Tregs generation and mechanisms of action, to focus, in the second part, on the effect of Tregs accumulation in sporadic CRC and CAC and on how Tregs may contrast or promote tumor growth and metastasis formation. Finally, we will give an interpretation of these data trying to understand whether and how Tregs targeting may represent a feasible therapeutic approach in patients affected by CRC.

## 2. Tregs Generation and Phenotype

Tregs are characterized by the expression of multiple surface markers, none of which specific, making difficult the isolation and study of Tregs ex vivo. So far, the most specific marker identified is FoxP3, a transcription factor which is essential to initiate and maintain Tregs suppressive phenotype [6,7]. Upon T-cell receptor T cell receptor (TCR) binding, in the presence of adequate environmental stimuli, FoxP3 is expressed in developing Tregs where it interacts with other transcription factors as NFATc [8], NFkB [9] and Runx1 [10], and regulates the expression of genes involved in Tregs functions. Loss of function mutations of *foxp3* cause autoimmunity in mice (*Scurfy*) [11,12] and in humans (Immune dysregulation, Polyendocrinopathy, Enteropathy, X-linked (IPEX) syndrome) [13]. Depending on the site of induction, Tregs have been classified in naturally occurring Tregs (nTregs), if induced in the thymus, or peripheral Tregs (pTregs) if generated in peripheral organs [14]. In the gut, Tregs are characterized by the co-expression of FoxP3 and the Th17-related transcription factor ROR t as a result of the interaction with the intestinal microbiota [15]. In both cases, TCR signaling is required together with other signals to induce a stable expression of FoxP3 [16]. The stability of FoxP3 expression is critical for Treg suppressive activity [17,18]. FoxP3 may be transiently induced in human conventional T cells after TCR stimulation or in murine naïve CD4+ T cells after TCR activation in the presence of transforming growth factor (TGF) 1 [19,20]. In “induced” Tregs (iTregs), FoxP3 expression is transient and rapidly lost after activation due to the persistent methylation of the Treg-specific demethylation region (TSDR) contained in the *foxp3* promoter. Indeed, fully suppressive and stable Tregs show an extended demethylation status of the *foxp3* promoter and a stable expression of FoxP3 [21]. A stable expression of FoxP3 is critical to skew the differentiation program toward a suppressive phenotype and the levels of FoxP3 fine tune Treg suppressive activity. High and persistent FoxP3 expression, as observed during inflammation, enhances the expression of Treg suppressive armamentarium as shown by FoxP3^high^ CD25^high^ CD45RO+ effector Tregs (eTregs) [22]. On the other hand, once FoxP3 expression is reduced or lost, Tregs acquire a “memory” phenotype or may transform in effector cells and secrete proinflammatory cytokines [23]. Whether tumor infiltrating Tregs are recruited from the nTreg pool or locally generated is unclear. Tumor associated macrophages (TAMs) and myeloid suppressor cells (MDSCs) release chemokines, such as CCL17, CCL22, CCL5, CCL6 or CCL28, depending on the tumor type, to attract Tregs expressing the chemokine receptors CCR4, CCR5, CCR10 and CXCR3 from secondary lymphoid tissues to the tumor [24]. Though, Tregs may be also induced locally by tolerogenic dendritic cells (DCs) [25] but whether this occurs in CRC is currently unclear. A common feature of Tregs infiltrating the TME is the effector phenotype characterized by high immune-suppressive capacity. Tumor-infiltrating Tregs from NSCLC, head and neck squamous cell carcinoma (HNSCC) and melanoma patients showed higher frequency and suppressive capacity as compared to Tregs from other sites [26,27,28]. The accumulation of eTregs in the tumor tissue may be the consequence of a strong antigen-specific TCR stimulation or the presence of co-stimuli provided by the specific TME. The effect of a prolonged TCR stimulation and FoxP3 expression includes the repression of *il2* gene and the activation of *il2ra* gene encoding for the IL2 receptor alpha subunit (CD25) that together with the IL2 receptor beta (CD122) and the common gamma-chain (CD132) subunits, forms the high affinity IL2 receptor. The high expression of CD25 is used by Tregs to deprive effector T cells from the proliferative and anti-apoptotic effect of IL2 acting as scavenger molecule (see below). In addition, high CD25 expression makes Tregs highly sensitive to low concentration of IL2 that through the activation of STAT5 co-operates for the stability of FoxP3 expression [29,30,31]. Moreover, FoxP3 interacts with other transcription factors such as Runx1, N-FATc2, NFkB and p300 to induce/enhance the expression of other Treg-related markers such as GITR, CTLA-4, PD-1, TIM-3, LAG-3, LAP and GARP. Co-stimulatory signals generated by GITR, OX40 and TNFR2 which activate NFkB, may also contribute to the maturation of Tregs into eTregs in the TME [32].

## 3. Tregs Mechanism of Action

So far, more than a dozen Tregs suppression mechanisms have been identified (Table 1). They can be divided in active suppression mechanisms that require the expression of immunomodulatory cytokines (e.g., IL10, TGF 1 and IL35) and counteractive mechanisms of suppression based on the active depletion of soluble and membrane bound molecules involved in effector cells activation and function [33] (Figure 1).

### 3.1. IL-10

IL-10 is expressed by immune and non-immune cells and activates the heterotetrameric IL-10 receptor composed by the IL-10 binding IL-10R1 homodimer and the signal transducer IL-10R2 homodimer. IL-10 inhibits phagocytosis, antigen presentation and the expression of costimulatory molecules. In T cells, IL-10 has been shown to reduce proliferation and to inhibit IL-2 and interferon (IFN)γ expression [34]. Of note, Tregs-specific IL-10 deficient mice develop gut inflammation and are more susceptible to proinflammatory challenges at the mucosal level, without recapitulating the multiorgan autoimmune phenotype observed in FoxP3 deficient mice [35]. These data indicate that Tregs immunosuppressive activity relies on multiple suppressive mechanisms and that IL-10 may have a prominent role in controlling tolerance against microbiota and dietary antigens [36]. IL-10 is secreted in the TME and it has been shown to induce an exhausted phenotype on tumor infiltrating T cells characterized by low proliferation and suppressed secretion of cytokines associated with an effector phenotype as IL-2, IFNγ and TNFα [37].

### 3.2. TGFβ

TGFβ is a family of cytokines involved in multiple biological processes targeting immune and non-immune cells whose prototype member is represented by TGFβ1. TGFβ1 signals by interacting with the TGFβ receptor complex formed by TGFβR1 and TGFβR2. TGFβ1 deficient mice or mice overexpressing the dominant negative form of the TGFβR complex develop autoimmunity [38]. In contrast to IL-10, that is actively secreted, TGFβ1 is exposed on Treg cell surface bound to latency associated protein (LAP) and Glycoprotein-A repetition predominant protein (GARP) which is responsible for the mobilization of TGFβ1 from the cytoplasm to the cell surface [39]. TGFβ1 release requires the mechanical dissociation or proteolytic cleavage of LAP by αV integrin heterodimers [40,41]. This mechanism might guarantee the selective release of TGFβ1 locally to antigen-specific effector T cells to suppress activation and to convert them into induced Tregs. Whether this phenomenon occurs via a *ménage à trois* with the participation of αV-expressing DCs or after the DC-mediated activation of TGFβ1 and a direct T-T interaction is currently unclear [42,43].

### 3.3. IL-35

IL-35 belongs to the IL-12 family of cytokines but, in contrast to IL-12 and IL-23, IL-35 displays immunosuppressive properties. IL-35 is a heterodimer cytokine composed by the Epstein-Barr virus-induced gene 3 (Ebi3) and the IL-12 p35 subunit that bind to the heterodimeric receptor formed by the IL-12R 2 and gp130 [44]. IL-35 deficient Tregs failed to protect mice from colitis in the adoptive transfer model [45]. IL-35 expressing Tregs have been shown to promote T cell exhaustion in the TME [46].

### 3.4. IL-2

Among counteractive immunosuppressive mechanisms, IL-2 depletion by Tregs has been long investigated. The role of IL-2 in T cell activation and effector T cell generation is well known. IL-2 is expressed after TCR engagement in the presence of a second activating stimulus provided by the interaction of CD28 with the co-stimulatory molecules CD80/CD86 expressed by antigen presenting cells (APC) [47]. Tregs express high levels of IL-2Rα to form the high affinity IL-2 receptor but they do not secrete IL-2. Therefore, the active depletion of IL-2 from the local microenvironment by Tregs expressing high levels of IL-2R is believed to reduce the availability of IL-2 for T cell activation [48]. Interestingly, Treg characterized by the deletion of CD25 and gain of function of STAT5, lose the capacity to suppress CD4+ but not CD8+ cells thereby indicating that CD25 is involved in Tregs-mediated suppression in a target cell-specific manner [46]. Taken together, these data suggest that high expression of CD25 in Tregs may act both as a “sink” to deprive effector cells of IL-2, thereby inducing anergy or apoptosis, and at same time as enhancer of Treg-mediated suppression.

### 3.5. Granzyme B and Perforin

Tregs express perforin and Granzyme B and a direct cell-mediated cytotoxicity against target cells contributes to the suppressive activity of these cells. In CRC, LAP+ Tregs expressed high levels of perforin and Granzyme B [49]. Although the release of these cytotoxic molecules by Tregs have been shown to kill tumor cells in other cancers, the evidence that this mechanism operates in CRC is still missing [50].

### 3.6. Extracellular ATP Depletion

Another suppressive mechanism based on the depletion of activating signals is represented by the conversion of extracellular ATP in adenosine. ATP is released in response to pathogen associated molecular patterns (PAMPs) through specialized channels or in an uncontrolled manner as in case of inflammation-induced tissue damage [51]. ATP represents a danger signal and it is able to induce inflammation and to activate CD4+ and CD8+ T cells [52,53]. Tregs are characterized by the high expression of the membrane ectonucleotidase CD39 and CD72 which inactivate AMP and generate adenosine [54,55]. Adenosine can induce immunosuppressive signals in DC and effector T cells which express high levels of the adenosine receptor A_2A_ [56]. CD39 expression by Tregs and adenosine released into the TME reduced the expression of ICAM-1 on endothelial cells limiting the accumulation of effector T cells in CRC [57].

### 3.7. CTLA4

CTLA4 is a membrane bound molecule which has been implicated in the cell contact-dependent suppressive activity mediated by Tregs. CTLA4 is expressed on the cell surface of effector cells early after activation while it is constitutively expressed by Tregs. CTLA4 and CD28 share the same ligands, the co-stimulatory molecules CD80/CD86 expressed by APC cells. The signal delivered by CD28 after TCR engagement is crucial for T cell activation. Indeed, based on the three-signals activation model, the cognate recognition of MHC/antigen and TCR complexes, the CD28 binding with CD80/CD86 costimulatory molecules and the IL2 autocrine secretion are required for fully activate T cells and prevent apoptosis and anergy [58]. CTLA4 has a high binding affinity for CD80/CD86 and it is able to outcompete CD28 thus limiting T cells activation [59]. In addition, upon binding of CD80/CD86, CTLTA4 reduces the availability of CD80/CD86 for CD28 binding by inducing trans-endocytosis [60]. Finally, CTLA4 has been shown to generate cell-intrinsic signals in T cells leading to the inhibition of T-cell activation/expansion [61]. CTLA4 is also able to generate a reverse signal through CD80/CD86 in APCs resulting in the induction of the enzyme indolamine-2,3-dioxygenase (IDO) [62,63]. IDO catabolizes the amino acid tryptophane in kynurenine. Kynurenine binds to aryl hydrocarbon receptors expressed on APC surface and induces a tolerogenic phenotype [64]. In addition, tryptophane depletion induces cell cycle arrest and apoptosis of T cells [65,66], increases the conversion of T cells into iTregs and stabilizes Treg phenotype [67].

### 3.8. LAG-3

Lymphocyte activation gene-3 (LAG-3) shows 20% homology with CD4 and it binds MHC class II on APCs with higher affinity than CD4. LAG-3 is expressed by many immune cells including NK cells, B cells, T cells and DCs [68]. LAG-3 is expressed by Tregs in the presence of effector cells and contribute to their suppressive activity [69]. LAG-3+ Tregs expand both into the tumor itself and in the peripheral blood of CRC patients. FoxP3+LAG-3+ isolated from these patients showed higher secretion of IL-10 and TGF 1 and suppressive activity than LAG-3-Tregs [70].

### 3.9. Co-Stimulatory Molecules “Stripping”

Recently, a new mechanism of suppression based on the depletion of activating molecules expressed by DCs has been proposed. This mechanism relies on the binding of MHC class II and co-stimulatory molecules and the “stripping” of these molecules from the DCs surface in a process similar to trogocytosis. The stripped DCs become unable to activate T cells and this mechanism has been shown to be antigen specific. Indeed, DCs presenting two antigens become unable to activate T cells with the same antigen specificity of the Tregs they interact with, but not T cells expressing a TCR specific for the second antigen [71]. However, that such a mechanism contributes to the suppression of the immune response against CRC still needs to be proven.

## 4. Sporadic and Colitis-Associated Colorectal Cancer

CRC is one of the third most common cancers in the world and the second most frequent cause of death in cancer affected patients [72]. More than 70% of CRC are sporadic, 5% of CRC develop in the context of genetic diseases (e.g., FAP) and the remaining 30% are associated with chronic inflammation as observed in inflammatory bowel disease (IBD) patients [5]. Sporadic and colitis-associated colorectal cancers show important pathogenic, morphologic and molecular differences. Sporadic CRC represents the end stage of a well-established sequence of genetic and epigenetic alterations sequentially involving different genes such as APC, KRAS and p53. In sporadic CRC, genetic changes are paralleled by the progressive appearance of morphological changes in the gut mucosa starting from aberrant crypt foci (ACFs), passing though intermediate and late adenomas to end with the development of carcinoma (i.e., adenoma-carcinoma sequence) [73]. In contrast to sporadic CRC, colitis-associated colorectal cancer (CAC) shows a distinct genetic and morphological development. Genes involved in CAC development partially overlap those of sporadic CRC but the sequence of their involvement is different. In CAC the sequence adenoma-carcinoma is not observed, and synchronous carcinoma foci can be observed in the context of chronically inflamed mucosa or associated with dysplasia-associated lesion or mass (DALM) [74]. The role played by inflammation in sporadic CRC and CAC is also different. Indeed, while in CAC chronic inflammation is the main driving force of tumor development [75], in sporadic CRC inflammatory cells accumulate in the tumor stroma due to bacterial translocation across a defective epithelial barrier [76,77]. Due to these important differences, the role of Tregs and their effect on the tumor-associated immune response will be analyzed separately.

### 4.1. Tregs in Sporadic CRC

Tregs are believed to suppress the immune response against tumor-associated antigens and the high frequency of Tregs in the tumor micro environment has been associated with poor or better prognosis depending on the type of cancer [78,79,80,81,82,83]. However, the role of Tregs accumulation in CRC stroma is unclear. In contrast to other types of cancers, initial studies on patients affected by CRC failed to identify any correlation between Treg and disease prognosis [84,85,86]. More recently, a positive correlation with earlier CRC stage and overall survival have been shown [87,88,89].

These apparently discordant data might result from the effects that Tregs cause in different areas of the tumor mass. Indeed, while the high frequency of Tregs in the center of the tumor is associated with higher pathological differentiation, no lymphatic invasion and lower TNM stage, the accumulation of Tregs at the invasive margin discriminated patients with shorter overall survival [86]. Since high density of cytotoxic CD8+ and IFNγ CD3+ T cells at the invasive margin is associated with better outcomes [90], the local accumulation of Tregs at this site might dampen the anti-tumor activity. In contrast, Tregs in the tumor mass are believed to negatively control tumor growth by inhibiting Th17 cells evoked by bacterial antigens. Defective mucin expression and tight junction organization occur early during tumor development. The presence of a “leaky” epithelial barrier allows the translocation of bacteria across the epithelial lining, leading to increased IL23 expression by tumor infiltrating macrophages, upregulation of IL17A and proliferation of dysplastic cells [76,91]. Therefore, the functional role of Tregs in the CRC stroma might be to limit tumor promoting inflammation (Figure 2).

The spatial distance between Tregs and CD8+ T cells is also relevant on the survival of sporadic CRC patients. Posselt et al. showed, in a cohort of patients affected by rectal cancer, that short distance between Tregs and CD8+ cells in the proximity of epithelial cells was associated with a poor prognosis, whereas the opposite was true in the stroma [92]. Though the invasive margin was not specifically investigated in this study, these data further indicate that different localization of Tregs within the tumor mass might lead to different outcomes. What remains unclear is whether Tregs acting in different areas respond to different antigens representing two distinct TCR specificity clusters, or whether Tregs specific for tumor antigens extend their suppressive activity to tumor promoting-, microbiota induced-inflammatory cells or vice versa.

Another possible explanation for the dual role of Tregs in sporadic CRC might come from the co-existence in the tumor tissue of different subsets of FoxP3-expressing cells with different functional phenotypes. Three fractions of FoxP3+CD4+ T cells have been identified in sporadic CRC on the basis of FoxP3 and CD45RA expression levels [93]. In addition to FoxP3^low^CD45RA+ naïve Tregs and FoxP3^high^CD45RA-effector Tregs, a third class of cells defined as FoxP3^low^CD45RA−, does not possess suppressive capacity and can secrete pro-inflammatory cytokines. High frequency of IL17A- and IFN-expressing cells were found among FoxP3^low^CD45RA− cells. Moreover, FoxP3^low^CD45RA− cells were characterized by *foxp3* low hypomethylation according with a low and unstable FoxP3 expression. These data indicate that highly suppressive eTregs and non-suppressive proinflammatory FoxP3-expressing cells may coexist in CRC TME. Noteworthy is the observation that among sporadic CRC with high FoxP3+ cells infiltration, high frequency of FoxP3^low^CD45RA− proinflammatory cells expressing IL12R and IFNγ was associated with a better overall survival [93]. These data suggest that FoxP3-expressing cells characterized by an incomplete epigenetic rearrangement fail to become fully functional Tregs and acquire a Th1-like phenotype, whereby contributing to the immune response against cancer cells. Whether FoxP3^low^CD45RA− represent a class of IFNγ expressing cells with distinct functions from conventional Th1 T cells or are the result of a defective differentiation program characterized by the transient expression of FoxP3, is currently unknown. However, loss of IFNγ expression in Tregs deficient for the Th1 lineage commitment transcription factor Tbet, showed a defective Th1 immune response in a mouse model of colitis thus suggesting that FoxP3+Tbet+ T cells expressing IFNγ might be instrumental for the generation of a Th1 immune response in the sporadic CRC context [94].

Another subset of Tregs observed in sporadic CRC is characterized by the secretion of IL17A. FoxP3+IL17A+ Tregs isolated from CRC specimens express the Th17-related transcription factor RORγt, the chemokine receptors CCR6 and CCR4 and the cytokines TGFβ and IL6 [95]. The origin of these cells is unclear. In vitro, IL17A is induced in Tregs by APCs in the presence of IL1β, IL6 and TGFβ and in hypoxic conditions [95,96,97]. In vivo, FoxP3+RORγt+ Tregs are enriched in the gut mucosa and they are believed to be induced by the gut microbiota [98]. Functionally, FoxP3+IL17A+ Tregs from sporadic CRC patients maintained their suppressive phenotype against CD4+ and CD8+ T cells [99] but they were involved in cancer initiation by targeting epithelial cells in an IL17A-dependent manner [95,100]. In APC^min/+^ mice, a murine model of familial adenomatous polyposis (FAP), loss of RORγt in Tregs caused a significant attenuation of polyposis thus involving these cells in CRC development [101].

These data support the concept that in sporadic CRC, a spectrum of Tregs with different functional properties may coexist and their phenotype, in addition to their localization in the tumor mass, should be considered to understand the net effect of Tregs on CRC development (Figure 3). However, more functional data in appropriate experimental models (e.g., selective spatial and/or phenotypic depletion of Tregs) are requested to consolidate this concept.

### 4.2. Tregs in IBD

Tregs are pivotal in maintaining immune homeostasis in the gut. Experimental models of colitis indicate that functional Tregs are required to maintain immune tolerance against antigens derived from gut microbiota or introduced with diet. Colitis induced by the adoptive transfer of CD4+CD45RB^high^ naïve T cells in Severe Combined Immunodeficiency (SCID) mice is prevented by the co-transfer of Tregs [102,103]. In this model, inflammation is caused by the immune response against antigens of the gut microbiota. Indeed, mice housed in germ-free conditions fail to develop colitis after the adoptive transfer of naïve T cells even in the absence of Tregs [104]. Moreover, colitis induced by the oral administration of destrane sodium sulphate (DSS) which induces epithelial cell damage and a dysregulated influx of antigens in the lamina propria, is attenuated by Tregs [105]. Therefore, Tregs appear to be the master controllers of T cell clones reactive against luminal antigens. Accordingly, defects of Tregs suppressive mechanisms as in the case of CTLA-4, IL-10, IL-35 or LAG-3 deficiency, are invariantly associated with the development of colitis [35,69,106,107].

In human inflammatory bowel disease (IBD), chronic inflammation of the gut develops due to an altered immune response against luminal antigens in genetically predisposed subjects. Though, the role of Tregs in the pathogenesis of IBD is unclear. Several studies failed to observe a reduction of Tregs in the lamina propria of IBD patients, indeed, Tregs resulted increased in inflamed as compared to non-inflamed areas [108,109,110]. Moreover, Tregs isolated from patients affected by either Crohn’s disease (CD) or ulcerative colitis (UC), were fully functional in vitro, arguing against an intrinsic defect of Tregs in IBD pathogenesis. These data support the hypothesis that resistance of target cells, rather than a defect in Treg suppressive capacity could contribute to inflammation in IBD patients. Indeed, CD4+ cells isolated from the lamina propria of CD patients, which overexpress Smad7, an intracellular inhibitor of the TGFβ signaling, but not those isolated from non-IBD patients, were resistant to suppression mediated by Tregs from the peripheral blood of healthy donors [111]. However, resistance of effector cells could be overcome by expanding Tregs from the peripheral blood of CD patients resembling effector Tregs (CD4+CD25+CD127lowCD45RA+) and characterized by stable phenotype and no secretion of pro-inflammatory cytokines [112].

### 4.3. Tregs in Colitis-Associated Colorectal Cancer

Data describing the role of Tregs in CAC in IBD patients are scarce and mostly derive from animal models. One of the most frequently used model of CAC is the azoxymethan (AOM)/DSS mouse model. In this model, mice are injected intraperitoneally with AOM, an alkilanting agent, followed by cycles of DSS given orally to mimic chronic relapsing remitting intestinal inflammation as observed in patients affected by IBD. These mice develop adenoma like lesions driven by intestinal chronic inflammation as observed in human IBD [113]. A recent study in IBD patients showed a reduced accumulation of FoxP3-expressing cells in CAC as compared to sporadic CRC. Though, similarly to sporadic CRC, patients with higher FoxP3+ cells in CAC tended to have a better prognosis [114]. Accordingly, in the AOM/DSS model, late stage CAC was associated with increased number of Tregs in the mesenteric lymph nodes and the antibody-mediated reduction of Tregs during the early stage of CAC development showed a reduced tumor progression associated with an increased frequency of activated T cells [115]. FoxP3+IL-17A+ cells were shown to accumulate in the colon of patients affected by ulcerative colitis and CAC [100]. The accumulation of FoxP3+IL-17A+ cells was CAC specific since they were not detected neither in melanoma nor in ovarian or renal cell carcinomas. These cells functionally suppressed T cells proliferation and IFN secretion. Though, FoxP3+IL-17A+ cells induced the expression of the proinflammatory cytokines IL-1β and IL-6 in autologous epithelial cells in an IL-17A-dependet manner. FoxP3+ cells co-expressing the IL-17A-related transcription factor RORγt were observed in the dysplastic areas of IBD patients [116]. Loss of RORγt restricted to Tregs was protective against CAC development in AOM/DSS treated mice and the small tumors developed in RORγt deficient mice showed low STAT3 activation and low proliferation of dysplastic cells. The loss of RORγt enhanced CTLA4 expression in Tregs and reduced IL6 secretion by CD11c+ DCs, activating a Treg-to-DC reverse signal dependent on FoxO3 [116]. Recently, pharmacological suppression of RORγt in FoxP3+RORγt+ Tregs isolated from the lamina propria of IBD patients prevented the destabilization of FoxP3 expression induced by proinflammatory cytokines and enhanced IL-10 secretion [117]. These data support the hypothesis that RORγt expression in Tregs confers the capacity to secrete proinflammatory cytokines that might be detrimental in the context of CAC. Though Tregs appear to play a specific role in CAC, most of functional data on Tregs in CAC are limited to in vivo experimental models, and data on IBD patients are urgently needed.

## 5. Manipulation of Tregs for the Therapy of CRC and CAC

Tregs are responsible, at least in part, for the inefficient activation of the immune system against tumor cells and depletion of Tregs is believed to be a valid therapeutic approach in many cancer types. Yet, sporadic CRC and CAC might represent a more challenging settings for the application of immunotherapy based on Treg-depletion. First, the role of Tregs in CRC, as discussed, is highly dependent on their site of action within the tumor tissue: suppressive against antitumor immune response at the invasion site, suppressive against the tumor promoting inflammation in the tumor stroma. Second, phenotypically different Tregs might co-exist in the tumor tissue at the same time and the net effect on tumor growth might depend on the balance among suppressive function, cytotoxic activity and cytokine patterns expressed by Tregs subsets. A selective depletion of Tregs in cancer is a major challenge. The use of anti-CD25 monoclonal antibody (PC61) to deplete Tregs has been widely used to investigate the effect of Tregs depletion in cancer models. However, the effect of anti-CD25 is not limited to Tregs, leading to a significant reduction of activated CD4+ and CD8+ effector cells generated during the immune response against cancer cells [118]. Daclizumab, a humanized anti-CD25 monoclonal antibody, depleted Tregs and effector cells in metastatic melanoma patients with no clinical benefit [119]. In breast cancer, Treg depletion by Daclizumab followed by vaccination with TAA induced antitumor immune response [120]. Recently, low dose Basiliximab, a monoclonal antibody directed against the IL-2 alpha chain, has been show to preferentially deplete CD25-expressing Tregs not affecting effector cells [121]. Even so, a therapeutic approach in patients affected by cancer based on the unspecific depletion of Tregs, might lead to relevant autoimmune side effects.

Modulation of Tregs function rather than cell depletion might have more chances of success in the therapy of CRC and CAC. For instance, pharmacological suppression of ROR t in Th17 and Tregs could reduce the expression of pro-tumorigenic cytokines in CRC and CAC patients. Suppression of RORγt in IL17A-expressing Tregs has been shown to reduce the expression of IL17A and IL22, two cytokines involved in CRC development leaving the suppressive capacity of Tregs unaltered [117].

An alternative strategy to modulate the activity of tumor-infiltrating Treg activity is represented by the neutralization of T-cell immunoglobulin mucin 3 (TIM-3). TIM-3 is a molecule expressed on the surface of terminally differentiated IFNγ-producing Th1 and CD8+ cytotoxic T cells. The binding of TIM-3 by galectin-9, a soluble s-type lectin, triggers an exhausted phenotype characterized by low proliferation and secretion of IL2 and IFNγ in tumor infiltrating CD4+ and CD8+ cells [122]. TIM-3 expression also characterizes Tregs infiltrating sporadic CRC and CAC. In T lymphocytes isolated from sporadic CRC patients, the co-blockade of TIM-3 and PD1, the target of the immune checkpoint inhibitor nivolumab and pembrolizumab, not only increased the frequency of IFNγ- and TNFα-secreting T cells and the proliferation of tumor antigen-specific CD8+ cells but also reduced the number of Tregs [123]. In the AOM/DSS model of CAC, TIM-3+ Tregs increased in the mesenteric lymph nodes (MLNs) and spleen in a disease stage dependent manner [115]. Activation of TIM-3 in Tregs enhanced the expression of molecules related to suppressive activity and TIM-3 is believed to be responsible, at least in part, for the higher suppressive capacity of tumor infiltrating Tregs as compared to Tregs isolated from peripheral blood [124]. Experiments in mice bearing CT26 colorectal cancer xenografts showed that the accumulation of TIM3+ Tregs precedes the accumulation of exhausted CD8+ and CD4+ T cells. TIM-3 blockade in tumor infiltrating Tregs induced the expression of IL10, a cytokines involved in T-cell exhaustion. Moreover, early, depletion of Tregs and neutralization of TIM-3 caused a significant and sustained tumor regression. These data suggest that TIM-3 neutralization alone or in combination with early Treg depletion might represent an effective strategy for the therapy of CRC and CAC.

Once given the possibility to reduce the suppressive activity of Tregs against CD8+ cytotoxic T cells, a key question would be who is the candidate patient to such therapeutic approach. CRC developing in patients affected by hereditary or sporadic mutations of genes encoding for DNA mismatch repair system are characterized by high microsatellite instability and high frequency of activated CD8+ cells into the TME [125,126]. This is probably due to the high frequency of non-self epitopes generated by the increased frequency of DNA mutations. In these patients high antitumor potential is kept in check by the high expression of suppressive molecules as CTLA4 and PDL1 by tumor cells and Tregs [127,128]. Encouraged by the high efficacity of immune check point inhibitors in mismatched repair system defective CRC, Tregs suppression and/or modulation of their activity could be effective in these category of patients [129].

## 6. Conclusions

At least two different immune responses coexist in CRC. The first, analogously to other cancers, is potentially generated by the immune response against TAAs. The second is elicited by antigens derived by the intestinal microbiota that pass through a functionally defective intestinal barrier due to the altered expression of tight junction proteins in dysplastic cells. In terms of tumor progression, the first type of immune response has the potential to counteract tumor growth and the metastatic process, the second promotes dysplastic cell proliferation and tumor progression by releasing a series of cytokines and factors in the TME. Based on this functional framework, the net effect of the immune system activation in the tumor context may be thought as the result of the balance between these two immune responses. In case the response to TAA prevails over microbiota-induced inflammation, the net effect of the immune system activation will result in a response against cancer. In the opposite scenario, where the tumor inflammation, sustained by an excessive exposure to luminal antigens dominates, or in conditions characterized by chronic inflammation as in the case of CAC, the pro-tumorigenic effect of the immune system activity may prevail. In this context, the suppressive activity mediated by Tregs will generate different outcomes depending on prevailing immune response and the depletion of Tregs might result in opposite effects depending on the individual immunologic status. Therefore, the understanding of this immunologic balance might be pivotal in forecasting the outcome of approaches based on the global reduction of Tregs in the tumor stroma.

Recent advances in the field of tumor immunology suggest that Tregs activity might be spatially and functionally compartmentalized. Tregs suppress different immune responses depending on their localization within the tumor. At the invasive margin, where the immune response against TAA mediated by cytotoxic CD8+ and IFN-producing Th1 cells prevails, Tregs might dampen the anti-tumor activity of the immune system. In contrast, in the tumor stroma, Tregs could negatively control the tumor-promoting inflammatory process. To design a strategy targeting Tregs in a site-specific manner would greatly enhance our capacity to break free the cytotoxic cells-mediated immune response against tumor keeping tumor-promoting inflammation in check.

Finally, the identification of signals involved in the modulation of Tregs suppressive activity or inducing specific T helper type-like phenotype (i.e., Th17-like vs Th1-like Tregs) might pave the way for a therapeutic approach based on the modulation of Treg functional activity.

In conclusion, Tregs play a multifaceted role in the immune response operating in both sporadic CRC and CAC and a therapeutic approach based on Tregs manipulation, alone or in combination with other strategies, should take into account of the uniqueness of this immune response with a personalized approach to each patient affected by CRC.

## Figures and Tables

**Figure 1 ijms-21-06744-f001:**
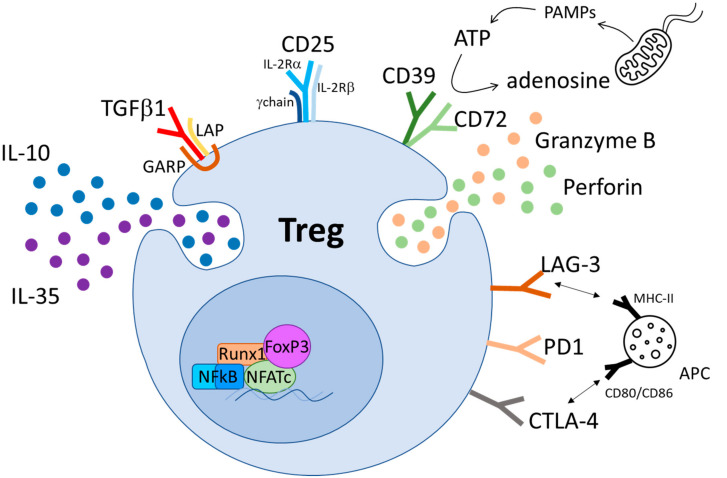
Treg-mediated mechanism of suppression. The FoxP3-containing transcription complex induces the expression of immunosuppressive cytokines (i.e., IL10, IL35 and TGFβ), cytotoxic molecules (i.e., Granzyme B and perforin) and surface molecules involved in Treg-mediated suppressive activity (i.e., CD25, CD39, CD72, LAG-3, PD1, CTLA4).

**Figure 2 ijms-21-06744-f002:**
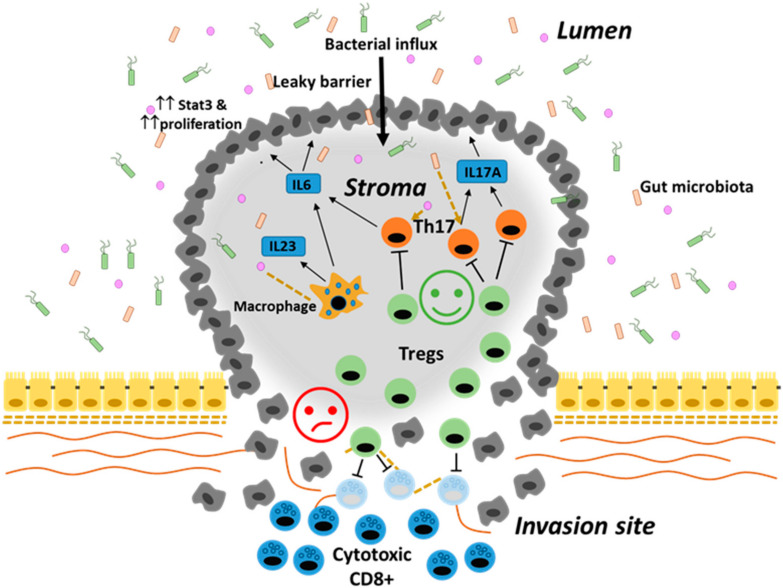
Dual role of Tregs in the TME: Tregs suppressive function against pro-tumorigenic inflammation evoked by bacterial antigens influx in the tumor stroma (smiling emoticon) and Tregs suppressive function against anti-tumor activity mediated by TAA-specific CD8+ cytotoxic cells (sad emoticon).

**Figure 3 ijms-21-06744-f003:**
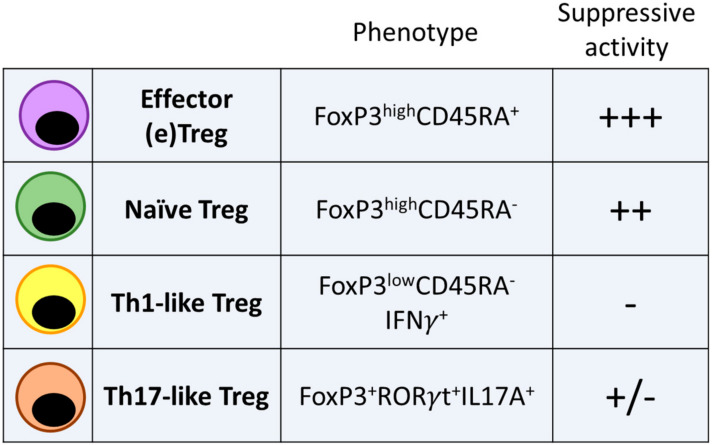
Phenotype and functional activity of Tregs subsets identified in sporadic CRC and CAC.

**Table 1 ijms-21-06744-t001:** Summary of molecule, mechanism of action and target cells operating in Tregs.

Molecule	Basic Mechanism of Action	Target Cells	Reference
IL-10	Inhibition of phagocytosis, antigen presentation and the expression of costimulatory molecules. Reduction of proliferation and IL-2 secretion. Induction of T cell exhaustion in the TME.	Antigen precenting cells (APCs), T cells	[34,37]
TGFβ	Suppression of T cell activation and conversion in Tregs. Suppression of proinflammatory cytokine secretion.	T cells and innate immune cells	[42,43]
IL-35	Inhibition of T cell proliferation and differentiation. Induction of T cell exhaustion in TME.	T cells	[44,45,46]
CD25	Induction of apoptosis and anergy in T cells by reducing IL-2 bioavailability in the TME	T cells	[47,48]
GanzymeB and Perforin	Direct killing of target cells	T cells, CRC cells (?)	[49,50]
CD39 and CD72	Conversion of extracellular ATP adenosine	Dendritic cells, T cells	[54,55,56,57]
CTLA4	Reduced T cell activation by outcompeting CD28 binding to CD80/CD86. Induction of IDO and activation of aryl hydrocarbon receptor signaling in target cells.	Dendritic cells, APCs	[58,59,60,61,62,63]
LAG-3	Suppression of target cell proliferation and effector cytokines secretion.	T cells	[65,66,67]

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
