# Peer review of "Tumor Infiltrating Regulatory T Cells in Sporadic and Colitis-Associated Colorectal Cancer: The Red Little Riding Hood and the Wolf"

_ijms, 2020, doi:10.3390/ijms21186744_

Round 1
Reviewer 1 Report
This is excellent review article titled "Tumor infiltrating Tregs in sporadic and colitis associated colorectal cancer: the red little riding hood or the wolf? " on the role of regulatory T cells (Tregs) in colorectal cancer and colitis-related CRC. I am fully supportive of its publication. A couple of minor suggestions:
- in the title, change Tregs to the full name.
- Discuss Tregs in the CRCs with mis-matched DNA repair gene deficiency, related to Lynch Syndrome or sporadic mutation of methylation of the gene promoter that result in the absence of expression of the mis-matched repair gene products leading to microsatellite instability, since this condition renders the CRC to be sensitive to immune checkpoint inhibitors.
- Discuss how co-stimulation with CD28 and CTLA regulate Tregs and CD4/CD8 T cells, for readers to understand more clearly the mechanisms that underlie the regulation of T cells by co-stimulation.
Thanks.
Reviewer 2 Report
- The title of the manuscript as a question is unclear.
- Please insert one or two sentences of conclusion at the end of the abstract.
- Lack of the list of abbreviations, or every abbreviation and it’s full name should be given at first time in the text.
- The clear description of all figure should be given, especially Figure 1.
- The figure No. 2 is not clear and readable, please improve it.
- The table is required at the end of Chapter 3:
molecule/ basic mechanism of action / target cells / number of references
- Separate chapter describing the role and importance of Tregs in the course of inflammatory bowel inflammation is very necessary.
- Please insert another chapters numbering:
4. Sporadic and colitis-associated colorectal cancer
4.1 Tregs in sporadic CRC
4.2 Tregs in colitis associated colorectal cancer
- Manipulation of Tregs for the therapy of CRC
- Summary
9. The conclusion concern with particular chapter should be given at the end of chapter 4.1 and 4.2.
10. Expression “CRC and CAC” is not correct, should be used “sporadic CRC and CAC”, because CRC does mean whole cancer group.
